# SCDBench: A Benchmark for LLM-Based Smart Contract Decompilers

## Abstract

Smart contracts are programs deployed on blockchains that manage digital assets and enable decentralized applications. While their bytecode is always accessible on-chain, more than 99% of Ethereum contracts lack verified source code, making decompilation essential for transparency and security analysis.

Traditional decompilers rely on program analysis to produce structured but low-level representations. Recent advances in large language models (LLMs) enable source-like output with higher readability and even recompilability. Yet systematic evaluation is missing: existing tools use narrow datasets and inconsistent metrics, hindering fair comparison and reproducibility.

We present the first systematic benchmark for smart contract decompilation. Our contributions are: (i) a diverse dataset of real-world contracts, filtered for redundancy and stratified by difficulty; (ii) a staged evaluation framework with metrics for format completeness, compilability, Application Binary Interface (ABI) recovery accuracy, and semantic equivalence; and (iii) baseline evaluations using a fine-tuned reference model, establishing a strong foundation for future research.

Our benchmark establishes a common ground for rigorous, reproducible evaluation and aims to accelerate the development of reliable smart contract decompilers for blockchain security and transparency.

## 1 Introduction

Smart contracts Wood et al. (2014) are programs deployed on blockchains that manage digital assets, enforce agreements, and underpin decentralized applications. They have transformed the blockchain ecosystem from simple value transfer to a programmable financial and computational infrastructure. Today, smart contracts secure over 160B USD across decentralized finance, non-fungible tokens, and governance systems.[1]

While the bytecode of smart contracts is publicly accessible on-chain, their source code is not always made available. Only a small fraction of contracts are voluntarily verified by developers and published in high-level languages such as Solidity. At the time of writing, more than 99% of the smart contracts on Ethereum, the largest smart contract-enabled blockchain, are unverified, leaving only low-level bytecode accessible.[2] This opacity hinders transparency, auditability, and accountability, as developers, users, and security researchers cannot easily understand or validate the behavior of deployed contracts. Recovering high-level representations from bytecode, i.e., decompilation, is therefore essential for security analysis and systematic understanding of the blockchain ecosystem.

Traditional smart contract decompilers Zhou et al. (2018); Grech et al. (2019; 2022); Lagouvardos et al. (2025) have relied on program analysis techniques, such as control-flow reconstruction and type inference. These tools often stop at producing structured intermediate representations (e.g., annotated pseudo-code) that are easier to follow than raw bytecode, but still challenging for developers to interpret. More recently, with the rise of large language models (LLMs), LLM-based decompilers have emerged. LLMs can greatly enhance readability and even generate source-like output that compiles back to bytecode, enabling correctness to be validated automatically. However, the field lacks a standardized benchmark. Existing tools are typically evaluated on proprietary or narrowly

---

[1]https://defillama.com/
[2]https://etherscan.io/contractsverified

scoped datasets, with inconsistent metrics, making it difficult to compare methods, reproduce results, or assess real-world effectiveness. As a result, there is no clear understanding of the relative strengths and limitations of different approaches.

In this paper, we present the first systematic benchmark for smart contract decompilation. Our contributions are threefold:

**Dataset.** We curate a dataset of contracts sampled from real-world deployments. To ensure diversity, we apply code similarity analysis to remove near-duplicates and capture a wide range of contract sizes and complexities. We further stratify contracts into subsets of varying difficulty levels, enabling fine-grained evaluation of decompilers under easy, medium, and challenging scenarios.

**Benchmarking methodology.** We design a staged evaluation framework with metrics including format completeness, compilability rate, Application Binary Interface (ABI) recovery accuracy, and semantic equivalence. These metrics collectively assess different aspects of decompiler quality, from structural validity to functional fidelity.

**Baseline evaluations.** We fine-tune a reference model and compare its performance against the corresponding base model. This benchmarked comparison establishes a strong baseline for future research. The results provide concrete insights into current limitations and offer valuable guidance for advancing the next generation of smart contract decompilers.

Our benchmark ultimately consists of a dataset of 150 contracts, covering 2,735 unique functions and corresponding to up to 27,350 differential fuzzing test cases. By establishing a common ground for evaluation, our benchmark aims to accelerate progress in smart contract decompilation. We believe it will foster reproducibility, enable fair comparison, and ultimately drive the development of more reliable tools for blockchain security and transparency.

## 2 RELATED WORK

There has been a growing line of research on smart contract decompilation, aiming to lift low-level Ethereum Virtual Machine (EVM) bytecode into more comprehensible high-level representations. Zhou et al. (2018) introduce Erays, which reconstructs control-flow graphs from EVM bytecode, lifts stack operations into a register-based form, and applies compiler-style optimizations to generate human-readable pseudocode. Grech et al. (2019) present Gigahorse, a declarative decompiler that translates bytecode into a three-address intermediate representation using Datalog rules for stack analysis, control-flow reconstruction, and function inference. Further advancing precision, Grech et al. (2022) propose Elipmoc, which extends Gigahorse with transactional context sensitivity and path-sensitive function reconstruction, enabling the recovery of private functions, arguments, and return values. Alongside these academic efforts, industry tools have emerged, such as Panoramix,[3] which relies on pattern matching, and Heimdall-rs,[4] which combines symbolic execution with decompilation. Complementing these decompiler designs, Liu et al. (2023) conduct a large-scale empirical study of five smart contract decompilers, systematically comparing their success rates, performance, ABI recovery, and resilience against compiler optimizations. Moreover, Lagouvardos et al. (2025) propose Shrnkr, a static-analysis-based decompiler that introduces shrinking context sensitivity and control-flow normalization, striking a balance between scalability and precision, and outperforming both static (Elipmoc) and symbolic (Heimdall-rs) approaches.

The advancement of LLMs have also opened new directions for smart contract decompilation. David et al. (2025) first fine-tune an LLM specifically for smart contract decompilation, using contracts lifted into structured three-address code to enable source-like Solidity recovery with improved readability. Su et al. (2025) present DiSCo, which forgoes additional model training and instead designs a frozen-LLM pipeline. DiSCo introduces semantic-unit intermediate representations, a type-aware graph neural network for variable name inference, and a prompt-synthesis framework that turns bytecode into structured natural-language descriptions.

Despite these advances, there remains a significant gap in how smart contract decompilers are evaluated. Prior evaluations of smart contract decompilers have mainly emphasized syntactic and structural aspects such as pseudocode readability and control-flow reconstruction. While these met-

---

[3]https://github.com/eveem-org/panoramix
[4]https://github.com/Jon-Becker/heimdall-rs

rics provide valuable insights, they are largely syntactic or structural in nature, and the evaluation datasets are often ad-hoc and not transparently documented. With the advent of LLM-based decompilers, the research focus is shifting toward compilability and semantic fidelity, which demand more rigorous and standardized evaluation. This motivates the need for a unified and transparent benchmarking methodology that can fairly compare emerging approaches. Among prior efforts, DiSCo's evaluation is closest in spirit to this goal with its use of explicit metrics, but the lack of transparency in dataset design and evaluation protocols prevents reproducibility and systematic comparison.

## 3  DATASET DETAILS

Although fewer than $1\%$ of deployed smart contracts on Ethereum are source-verified, this still amounts to more than 800,000 contracts with publicly available Solidity source code. These verified contracts provide a unique opportunity: they cover a wide range of application domains, coding styles, and levels of complexity, while also reflecting realistic distributions of compiler versions and optimization settings. We leverage this rich corpus to construct a benchmark dataset that balances realism, diversity, and analytical value.

### 3.1  DESIGN PRINCIPLES

Our dataset construction follows three complementary principles:

**Control redundancy.** Publicly verified contracts include large families of near-identical bytecode, for example ERC20 tokens generated from common templates. If such contracts are over-represented, they can artificially inflate evaluation metrics. To ensure fairness, we identify near-duplicates and retain only representative instances.

**Capture variety in difficulty.** While avoiding redundancy, we aim to cover contracts of different code lengths. Code size provides a practical proxy for decompilation difficulty: short contracts often compile into straightforward bytecode with simple control flow, whereas long contracts tend to encode multiple modules, libraries, or features, which are more challenging to decompile. By selecting across size ranges, we ensure that the dataset tests both basic and advanced capabilities.

**Ensure benchmark relevance.** The dataset must serve as a meaningful testbed: it should reflect realistic smart contract distributions while also containing challenging cases that stress-test decompilers. Our construction therefore balances representativeness with diversity, avoiding trivial collections of template contracts while including contracts of sufficient complexity.

### 3.2  DATASET CONSTRUCTION PIPELINE

The dataset is constructed in three stages: similarity fingerprinting, clustering, and controlled sampling. This pipeline consolidates near-duplicates into coherent families and then selects a representative yet diverse subset suitable for benchmarking decompilers.

**Similarity fingerprinting.** Each contract's runtime bytecode is first disassembled and normalized. Normalization masks immediates by collapsing all `PUSHk` opcodes into a single `PUSH#` token, coarsens `DUPk` and `SWAPk` to `DUP#`/`SWAP#`, and compresses consecutive `JUMPDEST` markers. From the resulting token stream, we extract opcode $n$-grams (default $n = 3$) and compute a 64-bit SimHash over the multiset of $n$-grams. SimHash is a locality-sensitive hashing scheme widely adopted in code similarity detection: contracts with similar features map to fingerprints within a small Hamming radius, while dissimilar contracts are well separated. We note that SimHash is approximate: different contracts can occasionally collide, and small code variations may sometimes lead to disproportionate distance changes. However, precise fingerprinting is not required for our purpose. Our aim is not to capture fine-grained semantic equivalence, but to cluster families of highly similar contracts (e.g., ERC20 templates, mass-deployed contracts) to prevent them from overwhelming the benchmark. For this task, SimHash provides the right trade-off between scalability and discriminatory power, while more exact methods (e.g., AST- or CFG-based similarity) would be significantly more expensive without materially affecting dataset diversity.

**Clustering.** Exact duplicates are merged by collapsing items that share identical 64-bit fingerprints. To detect near-duplicates, we apply a banded locality-sensitive hashing (LSH) scheme: each finger-

print is partitioned into bands, and items matching in at least one band are considered candidates. To avoid oversized buckets caused by mass reuse or bulk deployments, we deterministically partition large buckets into smaller sub-buckets using a fast mixer, and only compare items within each sub-bucket. Candidate pairs are verified via exact Hamming distance on the fingerprints, retaining edges at distance $\leq r$. The resulting similarity graph is partitioned into connected components (via union–find), yielding clusters that represent families of near-duplicate contracts.

**Controlled sampling.** Sampling is stratified by bytecode length, used as a coarse proxy for decompilation difficulty. We compute the empirical 33rd and 66th percentiles over all contracts, denoted $q_{\text{easy}}$ and $q_{\text{hard}}$. Contracts with length $L \leq q_{\text{easy}}$ are assigned to the easy bin, those with $L \geq q_{\text{hard}}$ to the hard bin, and the remainder to medium. Within each bin, we select $k$ contracts using a size-aware, diversity-seeking procedure. Let $s_c$ denote the size of cluster $c$. At each step, we choose a cluster with probability proportional to $s_c^{\alpha}$ ($\alpha = 0.5$ by default), using a round-based rotation so that a cluster is not revisited until all eligible clusters have had a chance. From the chosen cluster, we select the contract whose 64-bit SimHash is farthest in Hamming distance from the set already selected in that bin. This yields a breadth-first sample across clusters that reflects prevalence (via $s_c^{\alpha}$) while promoting dissimilarity among representatives, without explicit per-cluster caps.

This methodology efficiently consolidates near-duplicate families, reflects real-world prevalence without allowing any single family to dominate, and promotes internal diversity among selected representatives. The resulting dataset is balanced, diverse, and analytically meaningful—properties critical for a transparent and robust decompiler benchmark.

### 3.3 DATASET STATISTICS

We begin by collecting all smart contracts deployed on Ethereum before June 30, 2025, totaling 78,440,377 contracts. For each contract, we query Etherscan to check whether it is verified, and if so, fetch the corresponding source code. We restrict our dataset to Solidity contracts, discarding those written in other languages such as Vyper. We then deduplicate contracts with identical bytecode. This yields 749,870 unique pairs of bytecode and source code, which serve as the raw pool for our benchmark. Note that comments are removed from source code to eliminate non-functional text. For contract source code spanning multiple Solidity files, we flatten them into a single file.

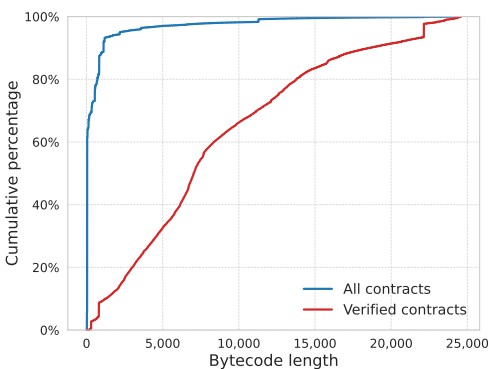
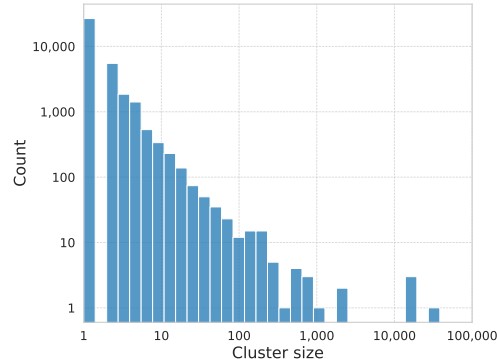

Figure 1: Cumulative distribution of bytecode lengths for all contracts and for verified contracts. The overall population is dominated by short contracts, whereas verified contracts are systematically larger and span a wide range of code sizes.

Figure 2: Histogram of cluster sizes on a log–log scale. Most clusters are very small, often consisting of a single contract, while a few extremely large clusters reach up to 584,973 contracts in the largest case, representing mass deployments of near-identical bytecode.

Figure 1 shows the cumulative distribution of bytecode lengths for all contracts and for verified contracts. A key reason for the sharp skew in the overall distribution is that more than 55% of all deployed contracts are Minimal Proxy Contracts,[5] extremely short bytecode fragments constructed directly without any associated source code. This explains why the majority of all contracts are

---

[5]https://eips.ethereum.org/EIPS/eip-1167

Table 1: Statistics of sampled contracts across difficulty bins.

| Difficulty | Count | Min | Max | Mean | Stdev |
|---|---|---|---|---|---|
| Easy | 50 | 209 | 5,028 | 2,088.42 | 1,394.50 |
| Medium | 50 | 5,072 | 9,801 | 7,128.96 | 1,366.59 |
| Hard | 50 | 9,955 | 24,468 | 15,072.78 | 4,381.66 |

very short, in contrast to verified contracts, which are systematically larger and exhibit a heavy tail. On average, the bytecode length of all contracts is $648.73 \pm 2,087.51$, whereas verified contracts average $8,618.68 \pm 6,244.50$. The distribution of verified contracts spans a wide range of code sizes, from compact utility libraries to complex protocols, providing a diverse basis for constructing a benchmark across different difficulty levels.

To avoid overwhelming the dataset with trivial clones, we apply the clustering methodology described in Section 3.2. Figure 2 presents the histogram of cluster sizes on a log–log scale. The distribution is highly skewed with a pronounced heavy tail: in total we obtain 36,905 clusters, of which 26,656 (72.23%) are singletons and 36,213 (98.12%) contain fewer than ten contracts. At the other extreme, the largest cluster contains 584,973 contracts, accounting for 78.01% of the entire corpus and reflecting mass deployments of ERC20-like templates. Between these extremes lies a long tail of medium-sized families, each with dozens to hundreds of contracts. This imbalance underscores the importance of redundancy control: without clustering, a handful of dominant templates would overwhelm the dataset and obscure the diversity of real-world smart contracts.

We then sample from the clusters to construct our benchmark dataset. Contracts are stratified by bytecode length into three difficulty levels: easy, medium, and hard. From each level, we select $k = 50$ representative contracts using the controlled sampling procedure described in Section 3.2, with $\alpha = 0.7$ to balance cluster prevalence against internal diversity. Table 1 summarizes the code size statistics of the resulting subsets. As intended, the easy bin is dominated by short contracts with simple control flow, whereas the hard bin contains substantially longer contracts that pose greater challenges for decompilation.

## 4 BENCHMARKS AND EVALUATIONS

### 4.1 METRICS

Evaluating smart contract decompilers requires metrics that capture both syntactic correctness and semantic fidelity. We define four progressive stages, ranging from basic format validity to full behavioral equivalence.

**Format completeness.** As the first stage, we check whether the decompiler produces a *complete, self-contained* output in the required format. This check is a necessary preparation for the second stage, where we test whether the output can be successfully recompiled. Concretely, the output must (i) specify the compiler version and settings, (ii) provide full Solidity code without ellipses, placeholders, or omitted dependencies, and (iii) state an unambiguous contract name. The compiler version and settings are required because Solidity syntax is *incompatible* across major versions (e.g., from v0.4 to v0.8), and successful recompilation depends on both the source code and the correct toolchain configuration. The contract name is necessary because a single Solidity file may define multiple contracts, for example through inheritance or libraries, each of which produces its own bytecode. Without an explicit contract name, it would be ambiguous which contract the decompiled output should correspond to when attempting recompilation.

**Compilability rate.** In the second stage, we evaluate whether the outputs that pass the format completeness check can be successfully recompiled. Using the declared compiler version and settings, we attempt to compile each decompiled contract. A successful compilation requires that the Solidity code is syntactically valid, consistent with the specified compiler configuration, and produces bytecode without errors. The compilability rate is defined as the fraction of decompiled contracts that compile successfully. This metric serves as a gateway check: only compilable outputs can proceed to subsequent stages such as ABI recovery accuracy and semantic equivalence.

**ABI recovery accuracy.** The third stage evaluates whether a decompiler can reconstruct the contract's external interface. We extract the ABI from the decompiled code and compare it against the ground-truth ABI derived from the original source. At the function-signature level, we measure precision, recall, and F1 score. This metric captures the decompiler's ability to recover callable functions and their prototypes, which are critical for interoperability, reverse engineering, and security auditing. Accurate ABI recovery ensures that users of the decompiled code can correctly understand and interact with the contract, even if deeper semantic fidelity is imperfect.

**Semantic equivalence.** The final stage evaluates whether the decompiled contract preserves the behavior of the original bytecode. We apply differential fuzzing: the decompiled output is recompiled, and both binaries are executed under a suite of automatically generated test inputs. For each input, we compare the returned outputs together with the resulting contract state changes, including revert behavior. A function is deemed equivalent only if all test inputs produce identical behavior; a single discrepancy marks it as non-equivalent. For each contract, we compute the ratio of equivalent functions to the total number of matched functions in the ABI. This metric provides the strongest validation: it goes beyond syntax and interface recovery to confirm that the decompiled contract faithfully reproduces the original program logic.

Together, these four stages form a progressive evaluation pipeline. Format completeness verifies that the output is well-formed and suitable for further testing; compilability checks basic syntactic validity; ABI recovery measures how accurately external interfaces are reconstructed; and semantic equivalence validates behavioral fidelity. We report all metrics across difficulty levels (easy, medium, hard) to provide a comprehensive assessment of decompiler performance.

## 4.2 REFERENCE MODEL

A core requirement of our benchmark is recompilation: decompiled outputs must form complete Solidity contracts that can be compiled back to bytecode. Traditional smart contract decompilers, however, typically stop at producing pseudocode or intermediate representations that improve readability but cannot be recompiled. David et al. (2025) decompile only at the function level, which rules out recompilation. DiSCo Su et al. (2025) does support compilability, but its implementation is not open source, preventing independent benchmarking. To enable a fair and open evaluation, we therefore decide to fine-tune a reference model that produces compilable Solidity contracts.

Our reference model takes inspiration from David et al. (2025), which fine-tunes language models for decompilation at the function level. In contrast, we target full-contract decompilation. This shift is made feasible by recent advances in LLM capacity, which allow us to bypass intermediate representations, and by longer context windows, which make it possible to process an entire contract at once. Specifically, we fine-tune `Qwen3-4B-Instruct-2507` as the base model. Figure 3 illustrates the prompt design. The model is instructed to decompile disassembled EVM bytecode and produce an output in a fixed structure consisting of three blocks: (i) a `<compiler>` block that

```
▤ System:
You are a specialized engine for decompiling EVM assembly into Solidity. Your
entire response must consist of a <compiler> block, a  block, and a
<contract_name> block, in that exact order.
```

```
👤 Human:
Decompile the following EVM assembly: PUSH1 0x80 PUSH1 0x40 MSTORE ...
```

```
🤖 AI:
<compiler>{\"compiler_version\": \"0.8.20\", \"settings\": ...} </compiler>
pragma solidity ^0.8.20;\n abstract contract Context {\n    function
_msgSender() internal view virtual returns(address) {...
<contract_name>UnibotLiquidityLocker</contract_name>
```

Figure 3: Prompts for the reference model.

Table 2: Tokenized sequence length statistics across difficulty levels, computed using the tokenizer of `Qwen3-4B-Instruct-2507`. The total sequence lengths are compared to the 32,000 token cutoff used in fine-tuning the reference model.

|  | Easy (50) | Medium (50) | Hard (50) |
|---|---|---|---|
| *Input length* | | | |
| Mean | 5,244.72 | 18,631.50 | 39,541.30 |
| Stdev | 3,365.67 | 3,644.42 | 12,506.08 |
| Min | 467 | 12,879 | 18,189 |
| Max | 13,525 | 27,458 | 67,371 |
| *Output length* | | | |
| Mean | 2,740.02 | 4,599.08 | 12,633.86 |
| Stdev | 4,434.66 | 3,205.27 | 12,112.22 |
| Min | 84 | 1,170 | 1,885 |
| Max | 29,585 | 22,678 | 73,673 |
| *Total length* | | | |
| Mean | 7,984.74 | 23,230.58 | 52,175.16 |
| Stdev | 6,230.59 | 4,891.09 | 19,526.95 |
| Min | 551 | 14,049 | 20,382 |
| Max | 36,443 | 36,450 | 109,120 |
| < 32,000 | 49/50 (98.00%) | 47/50 (94.00%) | 5/50 (10.00%) |

specifies the compiler version and settings, (ii) a `` block that contains the Solidity source code, and (iii) a `<contract_name>` block that indicates the contract name.

`Qwen3-4B-Instruct-2507` natively supports a context length of 256K tokens. For efficiency, we cap the maximum sequence length at 32,000. We fine-tune the model on 498,257 contracts from our verified corpus that fall within this length cap, explicitly excluding the 150 benchmark contracts used for evaluation. In addition to the fine-tuned reference model, we also benchmark the unmodified base model. This setup provides an open and reproducible baseline for our benchmark, enabling direct comparison with LLM-based decompilers.

## 4.3 EVALUATION SETUP

We run our evaluation on an NVIDIA RTX PRO 6000. Each decompilation attempt is limited to a timeout of five minutes. Due to the randomness of autoregressive generation, the model occasionally produces excessively long outputs that cannot finish within the time limit. In such cases, we retry the contract, for at most three attempts in total. If all attempts time out, we consider the model to have failed the first-stage evaluation of format completeness, since no complete output is generated.

For compilation and execution, we rely on the `foundry` toolkit.[6] All eligible outputs are recompiled with the compiler settings specified by the decompiler. For differential fuzzing, we generate 10 random test inputs for each recovered function. All executions are performed on a forked Ethereum mainnet state at block 22820673, so that contracts run against the real on-chain environment.

## 4.4 BENCHMARK DATASET CHARACTERISTICS

To better characterize our benchmark dataset, we analyze the tokenized sequence lengths of contracts across the three difficulty levels. Table 2 summarizes the tokenized input, output, and total sequence lengths across the three difficulty levels, where tokenization is performed using the tokenizer of `Qwen3-4B-Instruct-2507`. We report mean, standard deviation, minimum, and maximum values, as well as the fraction of contracts whose total sequence length falls below the 32,000 token cap used during fine-tuning. We notice that nearly all contracts in the easy and medium bins fit within this limit, whereas only a small fraction of hard contracts fall below it. Importantly, all contracts in our dataset remain within the 256K-token context window

---

[6]`https://github.com/foundry-rs/foundry`

Table 3: Format completeness results across difficulty levels.

| Difficulty | Reference model | | | Base model | | |
|---|---|---|---|---|---|---|
| | Timeouts | Wrong format | Eligible | Timeouts | Wrong format | Eligible |
| Easy | 6 (12%) | 0 (0%) | 44 (88%) | 2 (4%) | 22 (44%) | 26 (52%) |
| Medium | 8 (16%) | 0 (0%) | 42 (84%) | 1 (2%) | 25 (50%) | 24 (48%) |
| Hard | 29 (58%) | 0 (0%) | 21 (42%) | 3 (6%) | 15 (30%) | 32 (64%) |

supported by `Qwen3-4B-Instruct-2507`. Nevertheless, the statistics show that contracts in the hard bin pose a significant challenge for models with shorter context windows, such as `Meta-Llama-3-8B-Instruct` with an 8K-token limit, underscoring the need for larger context capacity in practical decompilation tasks.

## 4.5 EVALUATION RESULTS

**Format completeness.** Table 3 summarizes the outcomes of the format completeness check across the three difficulty levels. For each contract, we record whether the model output times out, produces an invalid format, or generates a complete decompilation eligible for recompilation.

The reference model shows markedly different behavior from the base model. After fine-tuning, it tends to generate longer (and probably more complete) outputs, with an average output length of 7,823.52 characters compared to 3,964.21 for the base model. These outputs are almost always structurally valid but more prone to exceeding the five-minute timeout. In contrast, the base model rarely times out but often produces incomplete or malformed outputs (e.g., invalid `<compiler>` blocks). Consequently, while the reference model suffers higher timeout rates, it achieves a 100% success rate on format validity whenever an output is produced, whereas the base model shows high rates of formatting errors, particularly in the easy and medium bins.

Table 4: Compilability results across difficulty levels. Numbers are successes and failures relative to the eligible set (from Table 3).

| Difficulty | Reference model | | Base model | |
|---|---|---|---|---|
| | Compiled | Failed | Compiled | Failed |
| Easy | 28/44 (63.64%) | 16/44 (36.36%) | 1/26 (3.85%) | 25/26 (96.15%) |
| Medium | 11/42 (26.19%) | 31/42 (73.81%) | 2/24 (8.33%) | 22/24 (91.67%) |
| Hard | 3/21 (14.29%) | 18/21 (85.71%) | 1/32 (3.13%) | 31/32 (96.88%) |

**Compilability rate.** Compilability is evaluated only on outputs that pass the format completeness check. Table 4 reports the number of contracts that successfully compile back to bytecode versus those that fail.

The reference model achieves substantially higher compilability rates than the base model in all difficulty levels, though absolute success rates decline sharply with contract size. In the easy bin, 63.64% of eligible outputs recompile successfully, while in the medium and hard bins the rates fall to 26.19% and 14.29%, respectively. By contrast, the base model rarely produces compilable outputs, succeeding on only a handful of cases across all bins. The weak performance of the reference model in the hard bin is likely explained by a training–evaluation mismatch: contracts exceeding 32,000 tokens were excluded from fine-tuning but remain present in the benchmark dataset.

A detailed breakdown of compilation errors is provided in Appendix A. The analysis shows that failures are dominated by declaration-related issues (e.g., redeclared or undeclared identifiers) and parser-level mistakes, while more complex semantic inconsistencies such as function signature collisions and inheritance errors occur less frequently. We also experiment with GPT-5 as a post-processing assistant: given the decompiled code and compiler error messages, GPT-5 repairs 61.54% of reference-model failures and 56.41% of base-model failures in a single zero-shot attempt, indicating the strong potential of LLM-assisted repair for improving compilability.

Table 5: ABI recovery results. Both macro- and micro-averaged precision, recall, and F1 are reported. Reference model substantially outperforms the base model across all bins.

| Difficulty | Macro average | | | Micro average | | |
|---|---|---|---|---|---|---|
| | Precision | Recall | F1 | Precision | Recall | F1 |
| **Reference model** | | | | | | |
| Easy | 0.873 | 0.845 | 0.854 | 0.915 | 0.932 | 0.923 |
| Medium | 0.991 | 0.987 | 0.989 | 0.992 | 0.989 | 0.991 |
| Hard | 0.987 | 0.987 | 0.987 | 0.988 | 0.988 | 0.988 |
| **Base model** | | | | | | |
| Easy | 0.417 | 0.714 | 0.526 | 0.417 | 0.714 | 0.526 |
| Medium | 0.306 | 0.588 | 0.401 | 0.259 | 0.538 | 0.350 |
| Hard | 0.455 | 0.625 | 0.526 | 0.455 | 0.625 | 0.526 |

**ABI recovery accuracy.** We compare recovered ABIs against ground truth at the function-signature level and report macro- and micro-averaged precision, recall, and F1 scores (cf. Table 5) The reference model achieves consistently high ABI recovery accuracy across all bins, with macro- and micro-F1 scores exceeding $0.85$ on the easy set and reaching near-perfect levels on medium and hard contracts. The base model, however, struggles: although it occasionally recovers partial interfaces, its F1 scores remain around $0.35$–$0.53$, reflecting both missing functions (low recall) and spurious predictions (low precision).

**Semantic equivalence.** Table 6 reports the ratio of equivalent functions per contract. Note that a function is deemed equivalent only if all test inputs produce identical behavior with the original bytecode (cf. Section 4.1). The reference model achieves mean ratios of $0.84$, $0.92$, and $0.92$ in the easy, medium, and hard bins, respectively. In the easy bin, $69\%$ of contracts reach perfect equivalence, while in the medium and hard bins this fraction drops to $18\%$ and $0\%$, despite higher mean ratios. By contrast, the base model shows consistently poor results, with mean ratios of $0.67$, $0.33$, and $0.08$ across the three bins and no contract achieving full equivalence. These results highlight the difficulty of attaining behavioral fidelity, while fine-tuning substantially improves equivalence rates, perfect preservation of logic remains rare, especially for more complex contracts.

Table 6: Semantic equivalence results across difficulty levels. Values report the ratio of equivalent functions per contract, averaged over contracts.

| Difficulty | Reference model | | | | Base model | | | |
|---|---|---|---|---|---|---|---|---|
| | Mean | Median | Stdev | % Perfect | Mean | Median | Stdev | % Perfect |
| Easy | 0.84 | 1.00 | 0.31 | 69% | 0.67 | 0.67 | 0.00 | 0% |
| Medium | 0.92 | 0.90 | 0.05 | 18% | 0.33 | 0.33 | 0.33 | 0% |
| Hard | 0.92 | 0.96 | 0.06 | 0% | 0.08 | 0.08 | 0.00 | 0% |

## 5 CONCLUSION

We present the first systematic benchmark for smart contract decompilation, combining a curated dataset of 150 contracts, a staged evaluation framework, and baseline evaluations with a fine-tuned reference model. The dataset balances redundancy control with diversity, spans multiple difficulty levels, and enables rigorous testing under realistic conditions. Our metrics, format completeness, compilability, ABI recovery accuracy, and semantic equivalence, form a progressive pipeline that reveals both syntactic and semantic strengths and weaknesses. Results show that fine-tuning greatly improves structural validity and interface recovery, but perfect semantic fidelity remains rare, especially for complex contracts. Post-processing with GPT-5 highlights the promise of LLM-assisted repair and suggests future multi-stage or agentic approaches.

This benchmark establishes a common ground for reproducible evaluation and aims to catalyze the development of more reliable smart contract decompilers, supporting greater transparency and security in blockchain ecosystems.

## REPRODUCIBILITY STATEMENT

We take reproducibility seriously and provide an anonymous repository (`https://anonymous.4open.science/r/SCDBench-5BD7/`) containing our benchmark dataset, together with the decompiled outputs of both the reference and base models. The repository further includes all scripts and instructions needed to reproduce our results on format completeness, compilability, ABI recovery, and differential fuzzing.

## LARGE LANGUAGE MODELS

We used GPT-5 to assist with editing and polishing of the manuscript text, including improving clarity, conciseness, and grammar. All research ideas, dataset construction, benchmark design, experiments, analyses, and conclusions were developed by the authors.

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

## A    COMPILATION ERROR ANALYSIS AND LLM-ASSISTED REPAIR

### A.1    COMPILATION ERRORS

To better understand why outputs fail to compile, we analyze the detailed error logs from the compilability evaluation. Table 7 summarizes the distribution of error types across difficulty levels. The majority of failures fall into a small set of recurring categories. *Declaration errors* are the most common, including redeclared or undeclared identifiers and name-resolution conflicts. *Parser errors* also occur frequently, such as "Expected primary expression" or "Expected type name," reflecting structural inconsistencies in the generated code. Less frequent but notable issues include type errors, invalid address literals, and inheritance-related conflicts. These results indicate that while many outputs are syntactically well-formed, a significant portion still encounter semantic or structural mismatches that prevent successful compilation.

Table 7: Breakdown of compilation errors across difficulty levels, with GPT-5 zero-shot post-repair outcomes. Each cell shows the number of compilation failures, with the number successfully repaired in parentheses.

| Error type | Reference model | | | Base model | | |
|---|---|---|---|---|---|---|
| | Easy | Medium | Hard | Easy | Medium | Hard |
| DeclarationError: Identifier already declared | 3 (0) | 5 (3) | 2 (1) | 4 (2) | 13 (9) | 17 (10) |
| DeclarationError: Undeclared identifier | 5 (4) | 9 (7) | 2 (1) | 3 (2) | 1 (0) | 2 (2) |
| ParserError: Expected primary expression | 0 (0) | 0 (0) | 1 (1) | 10 (6) | 2 (1) | 0 (0) |
| TypeError: Member not found / not visible | 0 (0) | 4 (4) | 1 (0) | 0 (0) | 1 (0) | 0 (0) |
| DeclarationError: Identifier not found or not unique | 1 (1) | 2 (0) | 2 (2) | 0 (0) | 0 (0) | 0 (0) |
| Error 8936: Identifier-start is not allowed at end of a number | 2 (2) | 2 (1) | 1 (0) | 0 (0) | 0 (0) | 0 (0) |
| ParserError: Expected type name | 0 (0) | 0 (0) | 0 (0) | 0 (0) | 2 (0) | 3 (1) |
| ParserError: Expected token | 0 (0) | 1 (0) | 2 (0) | 1 (0) | 1 (0) | 0 (0) |
| TypeError: Not implicitly convertible | 1 (1) | 0 (0) | 1 (0) | 0 (0) | 0 (0) | 1 (1) |
| SyntaxError: Invalid address literal checksum | 0 (0) | 1 (0) | 1 (1) | 1 (0) | 0 (0) | 0 (0) |
| Error 1860: Function signature hash collision | 0 (0) | 0 (0) | 0 (0) | 0 (0) | 1 (0) | 2 (0) |
| DeclarationError: Duplicate function signature | 0 (0) | 0 (0) | 0 (0) | 1 (1) | 0 (0) | 1 (1) |
| Error 2915: Expected a state variable declaration | 0 (0) | 0 (0) | 0 (0) | 2 (2) | 0 (0) | 0 (0) |
| TypeError: Free functions cannot have visibility | 0 (0) | 0 (0) | 0 (0) | 0 (0) | 0 (0) | 2 (1) |
| TypeError: Invalid type for argument (implicit conversion) | 0 (0) | 2 (1) | 0 (0) | 0 (0) | 0 (0) | 0 (0) |
| TypeError: Explicit type conversion not allowed | 0 (0) | 0 (0) | 1 (1) | 1 (1) | 0 (0) | 0 (0) |
| Error: Linearization of inheritance graph impossible | 1 (1) | 0 (0) | 0 (0) | 0 (0) | 0 (0) | 0 (0) |
| Error 4957: This type is only supported in ABI coder v2 | 0 (0) | 0 (0) | 1 (1) | 0 (0) | 0 (0) | 0 (0) |
| TypeError: Operator assignment type mismatch | 0 (0) | 1 (1) | 0 (0) | 0 (0) | 0 (0) | 0 (0) |
| Error 6480: Must override function | 0 (0) | 0 (0) | 1 (1) | 0 (0) | 0 (0) | 0 (0) |
| Error 8015: Invalid type for argument in the bytes | 1 (1) | 0 (0) | 0 (0) | 0 (0) | 0 (0) | 0 (0) |
| Error 1856: Literal or identifier expected | 0 (0) | 0 (0) | 0 (0) | 1 (0) | 0 (0) | 0 (0) |
| Error 7139 | 0 (0) | 0 (0) | 0 (0) | 1 (1) | 0 (0) | 0 (0) |
| Error 5883: Event with same name and parameter types defined twice | 0 (0) | 1 (0) | 0 (0) | 0 (0) | 0 (0) | 0 (0) |
| Error 1227: Index range access is only supported for dynamic calldata arrays | 0 (0) | 0 (0) | 0 (0) | 0 (0) | 1 (0) | 0 (0) |
| TypeError: Operator not compatible with types | 0 (0) | 1 (1) | 0 (0) | 0 (0) | 0 (0) | 0 (0) |
| Error 2973: Wrong argument count for modifier invocation | 0 (0) | 0 (0) | 1 (1) | 0 (0) | 0 (0) | 0 (0) |
| Error: Visibility already specified as "public" | 0 (0) | 1 (1) | 0 (0) | 0 (0) | 0 (0) | 0 (0) |
| DeclarationError: Contract should be abstract (missing implementation) | 1 (0) | 0 (0) | 0 (0) | 0 (0) | 0 (0) | 0 (0) |
| Error 2614: Indexed expression must be a type, mapping, or array | 0 (0) | 0 (0) | 0 (0) | 0 (0) | 0 (0) | 1 (1) |
| Others | 1 (0) | 1 (1) | 1 (0) | 0 (0) | 0 (0) | 2 (2) |

## A.2 LLM-ASSISTED REPAIR WITH GPT-5

We further experiment with using GPT-5 as a post-processing assistant to repair compilation failures. In this setup, GPT-5 receives both the decompiled Solidity code and the compiler error message, and is asked to return a corrected version. Table 7 summarizes the outcomes, showing for each error type the number of failing cases and, in parentheses, the number that compile successfully after a single zero-shot repair attempt.

For the reference model, there are 65 compilation failures in total, of which GPT-5 repairs 40 (61.54%). For the base model, there are 78 failures, of which 44 (56.41%) are repaired. The repaired cases include a large fraction of *declaration errors* and *parser errors*, such as undeclared identifiers or "Expected primary expression," which are often resolved with small edits. By contrast, repair rates are much lower for *semantic inconsistencies*, including function signature hash collisions, inheritance linearization errors, and missing function overrides, which require deeper reasoning about program structure.

It is important to note that this experiment is conducted in a zero-shot setting: GPT-5 is invoked once with the error message and produces a single revised program. A failure to repair therefore does not necessarily mean the model is incapable of fixing the reported error, as contracts may contain multiple simultaneous issues and resolving one may expose another. Nevertheless, these results indicate that even a single-pass LLM repair step substantially improves compilability, and highlight the potential of iterative repair strategies.

## A.3 DISCUSSION AND FUTURE DIRECTIONS

These experiments demonstrate that LLM-assisted repair can significantly improve compilability, even in a zero-shot setting. We believe that additional interaction rounds, where the model incrementally refines its output based on new diagnostics, would resolve an even greater share of errors. The same principle extends naturally to the later evaluation stages: differential fuzzing could gener-

ate counterexamples, which in turn can be fed back to an LLM to iteratively revise the decompiled output.

This suggests an important future research direction: *agentic pipelines* for decompilation. One agent produces the initial decompilation, a second agent repairs compilation issues, and a third agent uses differential fuzzing feedback to refine functional correctness. Such multi-agent, feedback-driven workflows could dramatically improve the reliability of smart contract decompilation. Developing these systems lies beyond the scope of this paper, but our results indicate their promise as a compelling direction for future work.

