# OpenReview forum: "SCDBench: A Benchmark for LLM-Based Smart Contract Decompilers"
_ICLR.cc/2026/Conference — Submitted to ICLR 2026_

### Official Review · Reviewer_pyGw · 2025-10-30

**Soundness:** 2
**Presentation:** 2
**Contribution:** 2
**Rating:** 2
**Confidence:** 4

**Summary:**

This paper is timely, well-written, and conceptually valuable, addressing an important gap — a standardized benchmark for LLM-based smart-contract decompilation. The staged evaluation (format → compile → ABI → semantic equivalence) is thoughtful and reproducible, and the anonymous repository and openness policy meet reproducibility expectations.

**Strengths:**

1. Smart-contract transparency and LLM decompilation are both emerging and important; this benchmark fills a genuine gap.
2. The four-stage design (format → compile → ABI → semantics) is systematic and easy to reproduce.
3. Writing, organization, and graphics are polished.

**Weaknesses:**

1. Novelty clarity: what is truly “first”?
You claim “the first systematic benchmark” but do not run widely-used non-LLM decompilers (Gigahorse, Shrnkr, Heimdall-rs, Panoramix) as external baselines on your dataset. Without those, it’s hard to judge whether the benchmark fairly stresses existing approaches and how LLM decompilers compare head-to-head. Add a baseline suite covering at least one static and one symbolic/industry tool and report all four stages on them.

2. Dataset Scale and Representativeness
The final benchmark includes only 150 contracts, corresponding to 2,735 unique functions. Although the construction pipeline (deduplication → clustering → stratified sampling) is sound, this sample size is insufficient for a benchmark claiming to evaluate semantic fidelity across the heterogeneous Ethereum ecosystem. Specifically: (1) With just 50 samples per difficulty bin, the variance across compiler versions, patterns, and application domains cannot be captured reliably; (2) The dataset omits Vyper, Yul, proxy upgrade patterns, and non-Ethereum EVMs. The verified-source bias means many prevalent contract archetypes.

3. Stage dependencies and denominators are unclear
Stage 3 (ABI) and Stage 4 (semantic equivalence) are defined as contingent on compilability, but Tables 5–6 report high scores in medium/hard while Table 4 shows very few compilable outputs (e.g., only 3/21 compiled in hard for the reference model). Always report N (eligible) / N (bin) alongside each metric, and show CIs or bootstrap intervals. Provide a small flow diagram per bin: “#total → #format-ok → #compiled → #ABI-scored → #fuzzed.”

4. Lack of Executable Source Code and Pipelines
The reference model fine-tuned from “Qwen3-4B-Instruct-2507” is not released in your repository, and 'weights' or 'checkpoints' are unavailable.

5. Fuzzing-based semantic equivalence needs much stronger methodology
Ten random inputs per function is insufficient to establish semantic equivalence for contracts with wide state spaces. Use coverage-guided fuzzing (e.g., Forge’s fuzz with coverage hooks) + boundary heuristics for calldata/value, different msg.sender, msg.value, reentrancies, and environment (block.timestamp/number). Include stateful sequences (multi-call traces), not only single invocations.

6. Missing strong baselines
Existing non-LLM decompilers like Gigahorse, Elipmoc, Heimdall-rs, Panoramix are cited but not quantitatively compared. Without them, the benchmark cannot demonstrate where LLMs truly add value.

**Questions:**

1. How do you guarantee that near-duplicate contracts or variants compiled from the same template do not appear across train/test splits?

2. Have you performed a global similarity clustering to prevent leakage, or only per-dataset filtering?

3. Beyond bytecode length, did you confirm that the “easy/medium/hard” bins correlate with actual decompilation complexity (e.g., control-flow depth, compiler optimizations, proxy patterns)?

4. How well does SCDBench represent DeFi, NFT, and infrastructure contract types? Are all Solidity versions (0.4–0.8) and compiler settings proportionally covered?

---

> ### Author Response · Authors · 2025-11-20
>
> R4: Missing comparisons to Gigahorse, Elipmoc, Heimdall-rs, Panoramix.
>
> We agree that existing decompilers such as Gigahorse, Elipmoc, Heimdall-rs, and Panoramix are valuable references. However, these tools do not output compilable Solidity, which makes it impossible to run Stages 2–4 in our pipeline. Even Stage 1 is not directly comparable, since their outputs target different intermediate representations rather than full-contract Solidity. We will clarify this point in the paper and explain why uniform evaluation across all stages is not feasible with current non-LLM tools.
>
> R4: Dataset representativeness (150 contracts, no Vyper/Yul, etc.).
>
> SCDBench focuses on Solidity because it is by far the dominant language for verified contracts on Ethereum, and thus the most practical target for a first benchmark. The 50/50/50 split reflects a balance between diversity and evaluation cost. We will clarify this rationale in the paper and note that extending the dataset to non-Solidity sources is a natural next step.
>
> R4: Stage-dependency reporting (N eligible per stage).
>
> Thank you for this suggestion. We will add denominators and per-bin flow diagrams to clarify how many outputs reach each stage.
>
> R4: Model weights not released.
>
> We will release the fine-tuned checkpoint in the final version.
>
> R4: Fuzzing methodology too weak (10 inputs).
>
> Our goal is consistency and reproducibility rather than full semantic coverage. We agree that coverage-guided fuzzing and stateful multi-call sequences are promising enhancements. Thank you for the suggestion. We will extend the experiments in the revision and clarify that SCDBench naturally supports stronger fuzzing strategies.
>
> R4: Similarity leakage across train/test.
>
> We thank the reviewer for raising this point. For the benchmark dataset, we perform full global clustering to ensure that no near-duplicate contracts appear within or across difficulty bins. For the training set, we only remove exact bytecode duplicates. We do not apply the full similarity-clustering pipeline to the training contracts because the goal of the reference model is simply to provide a reproducible baseline rather than to establish a strict separation between train and test distributions. We will clarify this distinction in the paper.
>
> R4: Correlation of easy/medium/hard with actual complexity.
>
> We will add an analysis showing that bytecode length correlates with several structural indicators of decompilation difficulty (e.g., control-flow-graph depth and number of basic blocks).
>
> R4: Coverage of contract types and Solidity versions.
>
> SCDBench is not designed to proportionally mirror the full on-chain distribution, but to provide a diverse and non-redundant sample spanning common contract families and compiler versions. Our clustering and stratified sampling naturally capture DeFi, NFT, and infrastructure patterns without enforcing exact proportionality. We will clarify this in the paper.

---

> > ### Comment · Reviewer_pyGw · 2025-11-28
> >
> > Summary:
> >
> > While the authors have addressed minor formatting issues, the admission in Point 6 (that the training set was not clustered against the test set) confirms a critical data leakage issue. A baseline model trained on near-duplicates of the test set serves no scientific purpose. Furthermore, refusing to compare against non-LLM tools on ABI recovery (Point 1) and defending weak fuzzing with 'consistency' (Point 5) undermines the paper's claim of being a rigorous, systematic benchmark.
> >
> > Detailed Comments:
> >
> > 1. On Missing Baselines (Gigahorse, Heimdall-rs, etc.)
> > While it is true that non-LLM tools cannot undergo recompilation (Stage 2) or differential fuzzing (Stage 4), they absolutely can be evaluated on Stage 3 (ABI Recovery). The authors must run the non-LLM tools for Stage 3 (ABI). Excluding them entirely limits the paper's scope to "LLM-only comparison" rather than a general "Decompilation Benchmark."
> >
> > 2. On Dataset Representativeness
> > The primary contribution of this paper is the benchmark. If the authors claim to "establish a common ground", they must absorb the computational cost to provide a statistically robust sample. 150 contracts is too small to capture the long tail of Ethereum contract logic.
> >
> > 3. On Stage-Dependency Reporting
> > Acknowledge the addition, but note that this likely confirms the "survivorship bias" flaw.
> >
> > 4. Model weights not released.
> > Why final version? However, verifying reproducibility is a core part of the review process. Postponing the release of checkpoints prevents the reviewers from verifying the baseline performance during the rebuttal phase.
> >
> > 5. On Weak Fuzzing
> > Calling a pass on 10 random inputs "Semantic Equivalence" is misleading. Random inputs rarely pass `require' gates (e.g., `onlyOwner', specific token balances). A "pass" here likely just means "both contracts reverted immediately." The metric measures "Revert Equivalence," not functional equivalence.
> >
> > 6. On Data Leakage (The Critical Issue in Paper)
> > The authors admit they only removed "exact bytecode duplicates" from the training set , while using fuzzy clustering (SimHash) to select the test set. This guarantees that "near-duplicates" of the test set exist in the training set. If the Reference Model was trained on the "cluster siblings" of the test set, its performance is a result of memorization, not generalization. A benchmark paper cannot present a baseline that cheats. Even if the goal is "reproducibility," a baseline with data leakage sets an artificially high bar that honest future models (which haven't seen the test clusters) cannot fairly compete against.

---

### Official Review · Reviewer_DFwe · 2025-10-31

**Soundness:** 3
**Presentation:** 3
**Contribution:** 3
**Rating:** 4
**Confidence:** 3

**Summary:**

This paper proposes SCD-Bench, a benchmark for evaluating the capability of large language models (LLMs) in de-compiling smart contracts to recover the original code. The authors curate a set of 150 smart contracts whose source code is publicly available to construct the benchmark. The authors then evaluate a base LLM with and without fine-tuning to demonstrate the effectiveness of the benchmark.

**Strengths:**

- The paper addresses an important problem, smart contract decompilation, which has not been extensively studied with LLMs. The problem is also well motivated (security and auditing of smart contracts).
- The writing is very well organized and easy to follow.
- The benchmark is well designed, with a clear methodology for selecting contracts and evaluating model performance.

**Weaknesses:**

- Limited number of models evaluated -- while it seems like the main focus of the works on the benchmark itself, a more extensive evaluation of different LLMs (coding/reasoning etc.) would strengthen the paper.
- The benchmark size is quite small (50 contracts per group). Can this really be considered as a representative benchmark for the task?
- There are some hyperparameter choices that are not fully justified -- e.g. 10 inputs? is there some sort of completeness guarantee? and why $k=50$ and why $\alpha=0.7$? More details on these choices would be helpful.

**Questions:**

- Was synthetic data considered for any part of the benchmark? Fine-tuning, additional evaluation data, etc.
- Why the choice Qwen-4B specifically? (other than context length) is there a reason?

---

> ### Author Response · Authors · 2025-11-20
>
> R3: Limited model diversity in evaluation.
>
> Our focus is the benchmark, not an exhaustive model comparison. We chose Qwen-3/4B for its long context and full reproducibility. We can provide additional baseline runs from other models in the revision.
>
> R3: Benchmark size (50 per tier).
>
> The 50/50/50 stratified sampling is a deliberate efficiency–coverage tradeoff: it covers many families while keeping evaluation time manageable for the community. The benchmark can be extended without modifying methodology.
>
> R3: Hyperparameter choices (10 fuzz inputs, etc.).
>
> Ten random calldata samples follow standard practice in differential testing and keep the evaluation tractable across 27k tests. The goal is comparative consistency, not formal completeness. We will clarify this and note that stronger fuzzing strategies can be plugged in.
>
> R3: Why Qwen-4B specifically?
>
> Two reasons: (1) long context window, which is critical for full-contract decompilation; (2) full reproducibility on commodity GPUs. This makes the benchmark accessible for future users.

---

### Official Review · Reviewer_KxDQ · 2025-11-01

**Soundness:** 4
**Presentation:** 4
**Contribution:** 3
**Rating:** 6
**Confidence:** 4

**Summary:**

This paper presents SCDBench, the first systematic benchmark for evaluating LLM-based smart contract decompilers. SCDBench consists of a curated dataset of 150 real-world Ethereum contracts and 2,735 unique functions with filtering redundant (e.g., template tokens like ERC20) and duplicate contracts. This paper also proposes a stated evaluation framework with metrics such as ABI accuracy and format completeness.

**Strengths:**

1. Important and interesting topics that provide insightful benchmarks for further research.
2. The benchmark smart contracts are meticulously selected. Duplication is a well-known issue in the world of smart contracts; I really appreciate the authors' efforts in removing duplicate and template contracts and providing a benchmark with high quality.
3. The benchmark also carries a solid baseline method based on finetuned LLMs.

**Weaknesses:**

1. The size of the dataset is somewhat limited. I would appreciate it if the authors could enlarge the dataset with more unique contracts in the future.

2. The evaluation mainly compares a single fine-tuned model to its base version.

**Questions:**

In general, this is a good benchmark paper with a rigorous and thoughtfully designed framework. I appreciate that the authors recognize the widespread duplication in real-world smart contracts and take concrete steps to remove redundant or trivial instances. I believe the paper introduced a high-quality and diverse dataset. My main concern lies with the baseline evaluation: the paper only reports results for a single fine-tuned LLM (Qwen3-4B-Instruct), without comparisons to other open-source models such as DeepSeek or Llama-based variants. Could the authors justify the reason for that?

---

> ### Author Response · Authors · 2025-11-20
>
> R2: Dataset size is limited.
>
> We agree that a larger benchmark is valuable. Our goal for SCDBench is to provide the first standardized, reproducible evaluation suite with transparent selection and metrics. Starting with 150 contracts keeps it practical for researchers to run full recompilation and semantic-equivalence tests. That said, the construction pipeline scales naturally, and we are happy to extend the dataset in the revision.
>
> R2: Only one fine-tuned model evaluated.
>
> The intent is to establish a clean baseline, not to compare models exhaustively. By training one reproducible reference model, we ensure that future work can measure progress consistently.
> We can add additional models in the revision to show that the benchmark generalizes.

---

### Official Review · Reviewer_YAt2 · 2025-11-01

**Soundness:** 3
**Presentation:** 3
**Contribution:** 3
**Rating:** 6
**Confidence:** 2

**Summary:**

The paper introduces the first systematic benchmark for smart contract decompilation, including a dataset of 150 real-world contracts and a staged evaluation framework with metrics for format completeness, compilability. The paper conducts baseline evaluations using a fine-tuned reference model, establishing a strong foundation for future research.

**Strengths:**

**Originality**
The paper introduced the first systematic benchmark for smart contract decompilation, with design principles that balance realism, diversity, and analytical value.

**Quality**
The paper constructed the dataset with staged pipeline that consolidates near-duplicates into coherent families and then selects a representative yet diverse subset suitable for benchmarking.

**Clarity**
The paper is well-written with description of benchmark construction method and characterstics. The benchmark is open-sourced on anonymous repo.

**Significance**
The paper proposed a systematic benchmark which can help researchers evaluate their approach of using LLM to decompile smart contracts.

**Weaknesses:**

- Dataset consisting of 150 smart contracts may still be small even though the authors try to get a representative yet diverse subset, there may be risk of overfitting.
- Evaluation only on a small finetuned LLM Qwen3-4b, so it's hard to know whether the benchmark can assess the capability of state-of-the-art LLMs.

**Questions:**

1. Why do you choose k = 50 to construct only 150 representative contracts of three different difficulty levels?
2. Why you choose to only evaluate on a small LLM instead of state-of-the-art LLMs?

---

> ### Author Response · Authors · 2025-11-20
>
> R1: Why choose k = 50, and is a 150-contract benchmark too small
>
> Our choice of 50 contracts per difficulty tier (150 total) reflects a practical tradeoff between coverage and evaluation efficiency. SCDBench is designed as a reproducible, unified benchmark, not a large-scale pretraining dataset. Increasing the size would raise compilation and differential-fuzzing cost, making the benchmark harder for the community to run. The 50/50/50 split already provides wide cluster coverage and difficulty diversity while keeping evaluation manageable (27k fuzz tests across 2,735 unique functions). The construction pipeline scales naturally, and we are happy to extend the dataset in the revision.
>
> R1: Why evaluate only a small model?
>
> The goal is to establish a baseline, not to exhaustively evaluate all LLMs. A smaller model (Qwen-3/4B) makes the setup fully reproducible. We agree larger models are interesting and can include additional results from stronger models in the revision.

---

### Meta-Review · Area_Chair_7tHG · 2026-01-08

**Summary:**

The article has one comprehensive review and is rated lowly at 2. The other reviews repeat the same two criticisms, but the authors did not add more data (may not have been possible) nor a new LLM (would be quite easy to do so) to convince the reviewers.

**Reviewer Concerns:**

YAt2
- The dataset size is small, and the evaluation is done on a single LLM. The authors offer increasing the size in a later revision (this is not clear when that would be).
- The authors do not offer more model results.

KxDQ
- Praises the article but notes the small size of the dataset.
- The single LLM evaluation is mentioned again.

DFwe
- Praises the writing but repeats the same size and single LLM criticism.

pyGw
- Praises the article for writing, design, and importance.
- Disagrees with the first benchmark claim.
- Repeats the small size criticism and notes the lack of competitor methods. The authors respond that the mentioned methods would not produce compilable code for comparison.
- Checkpoints and weights are not shared. The authors promise but do not release the weights.
- Raises doubts on performance and experimental setup. The authors clarify and defend their choices.

**Reviewer Scores:**

YAt2 rated 6 and would keep the rating.
KxDQ rated 6 and would keep the score.
DFwe rated 4 and would keep the score.
pYgw rated 2 and would likely keep its rating.

---

### Decision · Program_Chairs · 2026-01-26

Reject